# Infants Display Anticipatory Gaze During a Motor Contingency Paradigm

**DOI:** 10.3390/s25030844

**Published:** 2025-01-30

**Authors:** Marcelo R. Rosales, José Carlos Pulido, Carolee Winstein, Nina S. Bradley, Maja Matarić, Beth A. Smith

**Affiliations:** 1School of Health and Rehabilitation Sciences, The Ohio State University, Columbus, OH 43210, USA; 2Planning and Learning Group, Departamento de Informática, Universidad Carlos III de Madrid, 28903 Madrid, Spain; jcpulido@inf.uc3m.es; 3Division of Biokinesiology and Physical Therapy, University of Southern California, Los Angeles, CA 90033, USA; winstein@pt.usc.edu (C.W.); nbradley@pt.usc.edu (N.S.B.); 4Interaction Lab, Thomas Lord Department of Computer Science, University of Southern California, Los Angeles, CA 90007, USA; mataric@usc.edu; 5Division of Developmental-Behavioral Pediatrics, Children’s Hospital Los Angeles, Los Angeles, CA 90027, USA; 6Department of Pediatrics, Keck School of Medicine, University of Southern California, Los Angeles, CA 90027, USA; 7Developmental Neuroscience and Neurogenetics Program, The Saban Research Institute, Children’s Hospital Los Angeles, Los Angeles, CA 90027, USA

**Keywords:** infant, motor learning, eye-tracking, visual-motor behavior, contingency learning, socially assistive robots

## Abstract

Background: Examining visual behavior during a motor learning paradigm can enhance our understanding of how infants learn motor skills. The aim of this study was to determine if infants who learned a contingency visually anticipated the outcomes of their behavior. Methods: 15 infants (6–9 months of age) participated in a contingency learning paradigm. When an infant produced a right leg movement, a robot provided reinforcement by clapping. Three types of visual gaze events were identified: predictive, reactive, and not looking. An exploratory analysis examined the trends in visual-motor behavior that can be used to inform future questions and practices in contingency learning studies. Results: All classically defined learners visually anticipated robot activation at greater than random chance (W = 21; *p* = 0.028). Specifically, all but one learners displayed a distribution of gaze timing identified as predictive (skewness: 0.56–2.42) with the median timing preceding robot activation by 0.31 s (range: −0.40–0.18 s). Conclusions: Findings suggest that most learners displayed visual anticipation withing the first minutes of performing the paradigm. Further, the classical definition of learning a contingency paradigm in infants can be sharpened to further the design of contingency learning studies and advance the processes infants use to learn motor skills.

## 1. Introduction

In adult learning, multiple types of motor learning (i.e., implicit and explicit learning) and behaviors (e.g., verbal statements of learned concepts, differences in movement and visual patterns) have been described that pertain to the learning process [1]. For example, in Brooks et al. [2], adult learners of a novel discovery task first used an explicit learning strategy to acquire the general movement pattern and then through a more implicit learning strategy, refined the skill and developed the motor tactics. The explicit learning process was identified through spontaneous verbal reports triggered by knowledge of performance during practice, while the implicit learning process was identified through a gradual improvement in motor performance. By exploring both implicit and explicit learning mechanisms, the investigator gains a more complete understanding of the learning process.

In the infant contingency learning literature, the classical operational indicator of learning is based on repetitive performance of the reinforced behavior (see [3,4] as examples). With a contingency learning paradigm, infants are reinforced to produce a specific behavior, like kicking a leg [5], flexing and extending the knee [6], or decoupling the hip and knee angel [7]. Learning is defined by increasing the number of reinforced behaviors during a contingency period (i.e., a period where an infant’s behavior is contingent for activating the reinforcement) compared to an unreinforced period recorded prior to the contingency period. However, the increased repetition of a behavior may reflect learning or possibly motivated practice independent of learning. Although motor practice and exploration are important for motor learning in the long term [8], solely focusing on the frequency of behavior fails to recognize that infants can rapidly connect behaviors and the outcomes of those behaviors [9]. This position that the classical definition of learning fails to assess learning can be seen in several literary works such as Kelso [10], Kelso and Fuchs [11], and Zaadnoordijk et al. [12]. All of these works suggest looking for a quantifiable measure that suggest that the infant learned the association between their movements and the outcome for their movements.

The assessment of self-agency has been proposed to be achievable through the utilization of contingency paradigms, as established in prior works [10,11,12]. Self-agency, a conceptual construct, is defined as the perception of controlling external events [13,14,15] or the realization of causing an effect in the environment [10,11]. In the context of the contingency learning literature, this refers to an awareness that one’s behavior results in reinforcement. Scholars have postulated that the examination of visual behavior within these paradigms is instrumental in ascertaining infants’ manifestation of self-agency [12,16].

The purpose of this study was to examine if infants who met criteria commonly indicative of learning a contingency learning paradigm (see second paragraph of the introduction, and Section 2.4.1) demonstrated predictive gaze on an outcome that was linked to their leg movements. Predictive gaze is defined as looking at the area of interest prior to an event occurring. This operational definition has been used in several paradigms that examine object permanence and action perception [17,18] and is viewed as evidence that an infant has learned a concept. However, it is not clear if infants display these anticipatory looks when learning novel motor skills or how this visual motor behavior evolves over trials in the context of a contingency learning paradigm.

In this study, we used the behaviors of a socially assistive robot [19,20] to reinforce infant right leg movements. Our primary aim was to determine if infants who were classified as learners (using the classical method) of a contingency paradigm demonstrated predictive gaze. We tested two hypotheses: (1) that infants who were classified as learners of the paradigm would display a right-skewed distribution for onset of their gazes on their activation of the robot; and the median value for their gaze onsets on the robot would occur prior to the robot’s activations; and (2) that classically defined learners would exhibit predictive gaze (time their gazes to occur prior to the activation of the robot (i.e., the reinforcement)) more than random chance.

Because we anticipated considerable variability in gaze behavior for motor learning in the context of a contingency learning paradigm, we explored the behavioral differences between infants classified learners and non-learners and the common visual motor patterns observed. Given the recent debate pertaining to the classical definition of learning the paradigm [10,11], we anticipated that classically defined learners and non-learners would be more similar than not.

## 2. Materials and Methods

### 2.1. Participants

Seventeen infants with typical development were enrolled in the study. The caregivers received $40 (USD) for their infant’s participation. They were recruited from fliers, online postings, and word of mouth in the greater Los Angeles area between October 2021 and April 2023. The inclusion criteria were that the infant was born full term (>37 weeks gestational age) and was between the age of 6 and 9 months old at the time of collection. Infants with any known visual, hearing, or orthopedic impairments were excluded. Additionally, infants with a first (sibling or parent) or second degree relative (aunt, uncle, half sibling, or grandparent) diagnosed with autism were excluded. Lastly, infants were excluded from the data set if they scored lower than the fifth percentile on the Bayley Scales of Infant and Toddler Development (fourth edition) for the average of all three domains tested: Cognitive, Language, and Motor [21] or if they cried continuously for 1 min during the experiment.

This is a first step, exploratory research study (as opposed to confirmatory research). No sample size calculations were performed because no previous similar studies exist to provide effect sizes. The target sample size for the study was 15 participants. This was determined using samples from prior literature [3,5,7]. Data from two infants were excluded from the data set, yielding a final sample of 15 infants. Data for one infant were not included due to a technical error in our robot software. Data for another infant were excluded due to crying longer than 1 min during the data collection. Participant characteristics for our final sample are found in Table 1.

### 2.2. Procedures

The research was approved by the Institutional Review Board of the University of Southern California (HS-14-00911). A parent or legal guardian signed an informed consent form before their infant’s participation. Data were collected at Children’s Hospital Los Angeles. At each data collection session, an infant’s anthropometric measurements were obtained (thigh length and circumference, shank length and circumference, foot length and width, and weight), and their motor, cognitive, and language development were assessed using the Bayley Scales of Infant and Toddler Development (version 4), and a parent filled out the First Year Inventory (FYI version 3.1) [22]. In total, the assessment duration ranged from one hour to an hour and a half. Breaks and feedings in between assessments were allowed to prevent fussiness and fatigue.

The Bayley-4 is a standardized observational assessment of motor, cognitive, and language development of children between the ages of 16 days and 42 months that yields standardized scores, age equivalents and percentiles based on normative data [21]. The FYI v3.1 is a 69-item parent report questionnaire about infant behaviors that may indicate an elevated likelihood for later neurodevelopmental conditions such as autism [23]. The questionnaire generates risk scores on seven computed factors; these factors were developed from a large community sample of infants ranging from 6 to 17 months of age who were followed to age 3 to assess diagnostic and developmental outcomes [23]. The factors are: 1–communication, imitation, and play; 2–social attention and affective engagement; 3–sensory hyperresponsiveness; 4–sensory hyporesponsiveness; 5–self-regulation in daily routines; 6–sensory interest, repetitions, and seeking behaviors; and 7–motor coordination and milestones. Data from the FYI are not reported here; instead they were used for a companion study of infants at elevated likelihood for being diagnosed with autism [24].

Infants were supported in a custom-designed infant chair in front of an infant-sized humanoid socially assistive NAO robot (Aldebaran United Robotics Group) (Figure 1). The infant’s posture was stabilized by a cloth band at the trunk which allowed for unrestrained movement of the arms and the seat allowed for unconstrained movements of the legs. The participant’s caregiver was seated to the left of them, and a researcher was behind the infant’s chair (Figure 2). The parent was in view of the infant and the researcher was out of the infant’s view. Last, the room in which the collection took place was isolated from external sounds.

Participating infants engaged in a twelve-minute and ten second contingency paradigm: a 2-min baseline, a 10-s demonstration, an 8-min contingency condition, and a 2-min extinction phase. During the contingency condition, the infant had the ability to activate the robot by using right leg movements (i.e., the reinforcement). Movements of the right leg by the infants caused the robot to produce a clapping motion and a laughing sound. In addition, the robot would display colorful lights. Infant right leg movements were not reinforced during the baseline, demonstration, or extinction periods.

The contingency structure was designed from prior literature. According to the conceptual background with the paradigm, the baseline is used to quantify the average rate of movement the infant would produce without reinforcement. Then the demonstration is used to show the infant that the robot can perform actions. Next the contingency portion of the task is used to assess if the infant will learn that they can activate the robot more than their baseline movements. Last, the extinction period is used to see if the infant will continue to move at the contingency level, even though the robot has stopped activating.

During the contingency paradigm, infants wore a head-mounted eye tracker (Positive Science^®^) over the right eye and four wearable sensors (APDM Inc., Portland, OR, USA) that were placed at the distal end of the arms and legs (one for each limb). For this study, the wearable sensors were used to record acceleration at 128 Hz for all four limbs. Eye gaze was recorded at 30 frames per second using the two cameras on the head-mounted eye tracker: the scene camera located on the infant’s forehead and the pupil camera located over the right eye. Prior to the start of the recoding, a 5-point calibration was performed for the eye gaze tracker by placing a globe toy with spinning bright lights in 5 different locations in front of the robot. The 5-point calibration was performed at a distance of 1.5 m in front of the seated infant, and between the infant and the robot.

The infant-robot interaction system employed in this study was comprised of the NAO robot and infant-wearable sensors. This is the same system that was developed and used in Fitter et al. [25]. Briefly, the robot activated during the contingency phase if the infant produced an acceleration of greater than 3 m/s^2^ as measured by the right leg wearable sensor. For further detail, please refer to Fitter et al. [25].

Lastly, a camera was used to record the infant and robot continuously during data acquisition. The camera was placed to one side of the infant to record behavior from both the infant and robot from a lateral view. Prior to the start of the contingency paradigm and after the 5-point calibration, the spinning globe toy mentioned above was turned on and off 3 times and shown simultaneously to the external cameras and to the eye tracker to synchronize all video data post collection.

### 2.3. Data Processing

Following data collection, eye gaze data were imported into the software program, (Yarbus, Positive Science). Yarbus eye tracking software synchronizes the two cameras on the eye tracker. Once the cameras were synchronized, a single researcher (initials) selected the 5 points that the infant looked at during the 5-point calibration in order to calibrate the eye gaze trace. Prior literature reports that the Positive Science^®^ eye tracker has a radial error 4-degrees [26]. To account for this error, a graphic overlay of 4-degree radius was imported to represent the estimated area that the infant was looking at during the paradigm. We also estimated the accuracy of the eye gaze trace by taking 5 frames from the calibration (one for each target) and used an open sourced accuracy calculator [27] to determine the accuracy for the infants as a group. For all collections the acceptable level of accuracy was 4-degrees, and participants did not exceed this value. The eye gaze data had an average accuracy of 1.4 degrees (standard deviation = 0.90), which was within our target range.

To inspect the accuracy of eye-gaze calibration and quantify data loss due to blinking or closed eyes, we analyzed eye-tracking data using a custom MATLAB script. For each participant, the total number of frames with a pupil diameter of 0 (indicating blinks or closed eyes) was divided by the total number of recorded frames. This provided the proportion of lost data. On average, 9.2% of the data were lost due to blinking or participants shutting their eyes (mean = 0.092, SD = 0.098).

Once the eye gaze video was created in Yarbus, it was then synchronized with the recoding from the lateral view video using ELAN software (ELAN 5.8, Language Archive). The first frame where the globe toy flashed was used to synchronize the eye gaze data and the side view recoding. After the synchronized video file was created, a custom Python software (python 3.9) program identified each time the robot was activated according to the video data’s timeline using the accelerometry data from the wearable sensors and the robot. The timing for each robot activation was confirmed in a frame-by-frame analysis.

The frame-by-frame video analysis was performed by three trained individuals to identify the behaviors of interest. Visual behaviors that were identified included: total duration of looking using the start and end of each gaze on any part of the robot (regions of interest defined in Figure 3a, Appendix A) and each instance of predictive and reactive gaze on the infant’s activation of the robot’s upper half (region of interest defined in Figure 3b, Appendix A). The upper half was defined as all parts of the robot from the bottom of the torso to the top of the chair. This included the hands and arms of the robot. Predictive gazes [17] were defined as a visual fixation (3 or more frames (i.e., 0.4 s) of no eye movement) [28] on the robot prior to its activation and no earlier than 12 frames (i.e., 0.4 s) prior to the robot’s activation [29]. In instances when no predictive gaze occurred, a reactive gaze [17] was defined as a visual fixation on the robot during its activation (60 frames). Lastly, if no gaze occurred on the robot during an activation, behavior coders marked the occurrence as a non-robot look. Figure 4 shows the event timeline of a single robot activation, where the leg movement occurs, the robot turns on, and then the robot turns off.

Behavior coders also identified the behavioral state of each infant over the course of the study. Behavioral state was coded as sleeping, drowsy, alert, fussy, or crying [30]. Alert was defined as a behavioral state where the eyes are visibly open and not clenched tightly. Fussy was defined as moments where the eyes were visibly clenching and opening; and the infant was making brief distressing noises. Crying was defined as periods of time where the infant had their eye always clenched and was making a constant distressed noise. In addition, tears would come out of the eyes. Infants were not assessed if they were sleeping (defined as a state of having their eye consistently closed and not clenched) or drowsy (eye shutting for extended period and opening for a brief second). Group data are presented in Table 2 for baseline, contingency and extinction phases.

Individuals who coded the video data were trained on test data sets and had to reach a reliability above 80% before independently coding video data from this study. One third of the videos were double-coded to determine reliability. For this subset of data, percent of agreement for behavioral state was 95.0%, type of visual gaze was 87.5%, and time spent looking at the robot was 95.3%.

### 2.4. Data Reduction

#### 2.4.1. Determining Learning Based on Leg Movement and the Classical Definition of Learning

An infant was classified as having learned the association between their action and the robot activating if the reinforced behavior is produced 1.5 times more during the contingency phase compared to the baseline [31]. If an infant activated the robot above the individualized learning threshold in a two-minute moving window during the eight-minute contingency phase, they were classified as having learned the paradigm. Two criteria were applied to frame and count each right leg movement: (1) leg acceleration had to exceed 3.0 m/s^2^ (i.e., the robot activation threshold), and (2) movement had to occur more than 2 s after the last counted movement (i.e., the duration of a single robot activation) qualified as a potential activation of the robot. Both criteria had to be met for an infant to activate the robot.

#### 2.4.2. Timing of Gazes on Each Robot Activation and Duration of Looking

To calculate the latency to onset of a gaze directed to the robot region of interest (Figure 3b), the start time for each gaze was subtracted from the start time of the corresponding robot activation. Thus, gazes identified as predictive yielded negative values and gazes identified as reactive yielded positive values (see Figure 4).

Figure 3a ROI was used to calculate the total duration of looking at the robot during the contingency period. This is a separate variable from predictive and reactive gazes. Duration of looking was the total time spent looking in Figure 3a ROI.

#### 2.4.3. Proportion of Gazes

The total number of predictive, reactive, and non-robot looks were divided by the total number of times the infant activated the robot. This was performed to calculate the proportion of responses for each type of gaze.

### 2.5. Statistical Analysis

Non-parametric tests and descriptive statistics were used to test our hypotheses. Effect sizes were also calculated for each comparison using Pearson’s r with the following explanations for r: small effect was |r| > 0.1, moderate effect |r| > 0.3, and large effect |r| > 0.5 [32]. All computations for frequency and proportion for the three types of gaze classification, learning classification, and onset of gazes on robot’s activation were computed using custom MATLAB programs and exported to SPSS (v.27) for analysis.

As stated in the Introduction, the aim of the study was to determine if classically-defined learners demonstrated visual anticipation of the outcome for their right leg movements. We first hypothesized that infants classified as classically-defined learners would demonstrate a bias in the onset of their gazes toward the robot’s activation. Thus, the distribution for the onset of an infant’s gaze would be positively skewed, i.e., the onset of the gaze would occur more in the predictive gaze region of the distribution. Qualitatively, the distribution for the onset of gazes on the robot’s activations was plotted using a frequency graph and these plots were visually inspected for skewness [33]. Quantitatively, we calculated skewness and the median for the onset of an infant’s gaze on their activations of the robot. Skewness values greater than 0.5 were considered positively skewed [33] while negative median values denoted that the center value for the timing of gazes preceded the robot activations.

Our second hypothesis aimed to determine if infants who were classically-defined learners displayed predictive gaze above the random chance level. Random chance would predict the 3 gaze behaviors as having an equal probability of occurring (i.e., 33.3%). The proportion of predictive and reactive gazes and non-looks were tested to determine if predictive gazes were occurring more or less than random chance using a two-sided one-sample Wilcoxon Signed-Ranked test for classically defined learners. No statistical adjustment was used for these analyses because our hypothesis was only concerned with predictive gaze behavior.

Lastly, several variables were compared to test for behavioral differences between infants classically defined learners and non-learners. A Wilcoxon Rank Sum test with Bonferroni adjustment was used to test for differences for the following variables: number of potential activations of the robot during the baseline and extinction phases (i.e., instances the infant would have activated the robot in response to their movements), total number of activations during the contingency phase, proportions for the three types of gazes, total time looking at the robot during the contingency phase, median gaze onset on the activation of the robot, skewness for the gaze onset on the activation of the robot, and average intertrial duration (i.e., the average time to activate the robot another time). Additionally, we used Spearman correlations to determine if being classically defined as a learner or non-learner was associated with percentiles scores from the Bayley-4 scales (cognitive, language, and motor scales). This analysis was used to assess for differences between the learners and non-learners and supported our exploratory analysis, described next.

## 3. Results

### 3.1. Number of Classically-Defined Learners Among Participants

Six out of 15 infants were classically defined as learners and 9 out of 15 infants were classically defined as non-learners (Table 3). Eye gaze data for two infants in the non-learner group were not collected due to intolerance of the eye tracker (see asterisked values in Table 3). However, these infants successfully participated in the data collection without the eye tracker and their data are included for all non-gaze related variables.

Additionally, no statically significant associations were found between learning classification and cognitive (r = 0.28, *p* = 0.31), language (r = −0.24, *p* = 0.39), and motor percentiles (r = 0.28, *p* = 0.30) using Spearman’s correlations.

### 3.2. Distribution for the Onset of Gaze for Classically-Defined Learners’ Activations of the Robot

In support of our first hypothesis, qualitative analysis confirmed that the distributions for gaze onset were skewed and trended to precede or occur with the onset of the robot’s response (Figure 5). The range for skewness fell between 0.56–2.42, with a median of 1.09. In real time, the onset of gaze ranged from −0.4 s (preceding robot activation) to 0.18 s (after the robot activation began), (median: −0.31 s).

### 3.3. Proportion of Predictive Gazes Among Learners

In support of the second hypothesis, a two-tailed one-sample Wilcoxon Signed Ranked test showed that classically defined learners visually anticipated the robot’s activations significantly higher than random chance (i.e., 0.333) (W = 21, *p* = 0.028) with a large effect (Pearson’s r = 0.895) (Figure 6). The proportion of reactive gazes (W = 12, *p* = 0.753, small effect- Pearson’s r = 0.128) and non-robot looks (W = 3, *p* = 0.116, large effect- Pearson’s r = 0.639) was not significantly different from random chance for classically-defined learners.

### 3.4. Exploratory Analysis Comparing Classically-Defined Learners and Non-Learners for All Variables

For this section we compared the classically-defined learners and non-learners following the order of our hypotheses, and then an overall examination of our remaining secondary behavioral variables. Comparing classically-defined learners and non-leaners, Wilcoxon Ranked Sum test showed no differences between the skewness (W = 53, *p* = 0.628, small effect-Pearson’s r = 0.16) and the median onset for the onset of gazes on the activation of the robot (W = 45.5, *p* = 0.628, small effect-Pearson’s r = 0.14). Results showed that classically defined non-learners had a range of skewness from 0.1–2.07 s (median = 1.61). The median for the median (min-max) gaze onset for the activation of the robot was −0.37 s (−0.4–0.75 s) for the classically defined non-learners (Figure 5).

Compared to classically-defined learners, classically-defined non-learners did not exhibit significant differences in the proportions of predictive gazes (W = 50, *p* = 1, no effect-Pearson’s r = 0.04), reactive gazes (W = 50, *p* = 1, no effect-Pearson’s r = 0.04), or non-robot looks (W = 52, *p* = 0.731, small effect-Pearson’s r = 0.12).

There was a statistically significant difference between groups with non-learners demonstrating a higher total number of movements that would have activated the robot during the baseline phase (W = 98, *p* = 0.0012, large effect-Pearson’s r = 0.79) compared with learners. There were no other differences between groups that were statistically reliable for the other secondary variable (Figure 7 and Table 4).

### 3.5. Exploratory Analysis for Behavioral Patterns of Gaze

As stated in the Introduction, we had a reasonable suspicion that infants classically defined as learners and non-learners would be more similar than not. Given this suspicion and the results presented, we further explored the visual motor data over time to see if any common behavioral trends were observed among the group as a whole. This analysis was performed to describe the patterns of visual motor behavior observed while the infants were participating in a contingency paradigm and to guide future contingency learning studies and to formulate new research questions about infant motor learning.

We plotted the timing and type of visual gazes on the robot activations (see Appendix A); as well as the total time spent looking in minute-by-minute blocks during the contingency phase of the paradigm (see Appendix A). We found that all infants exhibited a time point where they displayed predictive gaze for the majority of reinforcements in one block for more than 5 activations (referred to as the predictive block from here on out; see Appendix A). Eleven of the thirteen infants showed a pattern where they increased the number of robot activations after the predictive block was identified (see Appendix A). The other two infants did not exhibit this increase in robot activations after the predictive block since their predictive block was identified in the previous block of time.

The next pattern that we observed was the trend in the other types of gazes that occurred after the predictive block occurred. Of the 11 infants who had data after the predictive block, all displayed a greater occurrence of reactive and non-robot looks after the predictive block than before. For example, in infant L4^CR^ (see Appendix A), the predictive block occurred in minute 3. Then, the infant exhibited a greater number of reactive and non-looks as the paradigm progressed compared to before minute 3.

Lastly, to explore the relationship between the time point where predictive gaze occurred for the majority of the time and the amount of time spent looking at the robot, we plotted looking duration in minute-by-minute blocks (Appendix A). Seven of the 13 infants decreased the amount of time spent looking as the contingency paradigm progressed (see Appendix A). Four infants maintained the amount of time spent looking at the robot after the predictive block. The last 2 infants did not have an additional block to compare after the predictive block, as noted above.

## 4. Discussion

The aim of the study was to determine if infants classically defined as learners of a contingency paradigm demonstrated visual anticipation while engaged with the paradigm. We found that learners visually anticipated their activations of the robot at greater than random chance (>33.33%). Additionally, the descriptive data support that classically defined learners were often timing their gaze to occur either close to or prior to activation of the robot. Therefore, data from the six classically-defined learners demonstrate visual anticipation while engaged in the paradigm.

Importantly, we do not conclude that only classically-defined learners displayed visual anticipation and demonstrated learning. This is because classically defined non-learners (n = 7 with visual behavior data), compared to classically defined learners, exhibited similar visual motor behaviors while engaged in the paradigm. In fact, the only variable that was significantly different between the two groups was the total number of leg movements that would have produced activations of the robot during baseline. Compared to infants who were classically-defined learners, classically-defined non-learners tended to move more in the baseline condition, resulting in a higher learning threshold (i.e., 1.5× baseline) compared to learners that was presumably more difficult to achieve. Thus, our results support that the classic definition of learning a contingency paradigm is insufficient at classifying learning within a single session.

Our exploratory analysis suggests that the use of the 1.5× learning threshold most likely classifies learning incorrectly. Results show that the learners and non-learners were behaving similarly in the contingency portion of the paradigm when there was a contingency to be learned. However, non-learners moved more during the baseline period, which made them more difficult to be defined as learners. In future studies, using a retention or transfer paradigm would be preferred to classify learners and non-learners; a researcher would be able to compare a performance period where the process of learning is evolving (in this paradigm that would be the contingency period) to a testing period where the infant is actively showing what they have learned or transferring what they have learned to a new situation. For example, in our study, the majority of infants were activating the robot most frequently towards the end of the contingency period. If an infant were to be re-tested in the same contingency paradigm the next day and moved their right leg at a similar frequency at the start of the paradigm on the following day as at the end of the previous day, this would support the theory the infant retained at least an implicit understanding that their behavior activated the robot. Additionally, if we were to reinforce an arm movement instead of a right leg movement in such a follow-up assessment, then we could assess whether the infant transferred or generalized learning from the leg movement experience to the acquisition of a new motor skill, i.e., that arm movement also activates the robot.

In our analysis of the entire sample (n = 15) and our detailed description of minute-by-minute individual data, we found that the infants displayed visual anticipation more than 33% of the time while engaged in the paradigm. Additionally, all 6–9-month-old infants showed a period of time where they were visually anticipating the outcomes of their behavior (referred to as the first predictive block). These individual data suggest that all 13 infants with complete gaze behavior data demonstrate a time within the 8-min contingency phase where they have learned the connection between their behavior and outcome of that behavior—a demonstration of self-agency. Following this demonstration, the infants continued to increase the production of the reinforced behavior, while at the same time exhibiting variable types of gazes and decreasing the amount of time spent looking at the robot. One viable interpretation of this observation is that the predictive gaze occurring for the majority of that time epoch reflects that infants have learned a construct or concept [18].

Our findings of distinct visual motor patterns may be related to the two behavioral processes found in adult discovery learning. Discovery learning is the process of acquiring a movement pattern necessary to achieve the goal for a novel motor task. The acquisition process involves learning the motor strategy of “what” serial pattern of movements to make and the tactics of “how” to scale one’s movements in order to be successful [2]. Participants are usually tasked with the challenge of manipulating a novel tool in order to move a cursor to a target [2] or pick up an object [34]. As such, participants must first search for possible solutions (i.e., the motor strategy or “what” movement pattern) and then once found, exploit and refine the correct motor strategy (i.e., learn the motor tactics or “how” to scale the movements) to become skilled at accomplishing the goal [2,34].

We suggest that our results show two phases of contingency learning through visual motor behavior, similar to those described by Sailer and colleagues [35]. The first phase is an exploratory phase where infants are using visual attention (i.e., behaviors considered to involve higher amounts of cognitive effort) to discover what their behaviors can produce in the world. We contend that all of our infants demonstrated this visual attention phase through visually anticipating more than 5 robot activations during a single 1-min block during the contingency phase.

Second, there is a skill refinement phase where infants are increasing their learned behaviors. We contend that based on our exploratory analyses of visual motor behavarior, eleven of our infants demonstrated this pattern. Specifically, after infants display prediction of their activation of the robot for the majority of the activations, they increase their production of the reinforced behavior while looking elsewhere.

While further assessment of these proposed learning phases of infant contingency learning are warranted, we suggest that they are plausible given our findings and past literature from adult motor learning [1,36,37]. If the proposed learning phases are further supported, it potentially indicates that when an individual is trying to teach a behavior to an infant, they should focus on establishing a behavioral connection during the exploratory phases of learning (i.e., helping the infant understand general movement pattern through visual cueing, modeling, and eliciting the behavior of interest) and then focus on repetition and variability in the refinement phases of learning the movement (i.e., encouraging the infant to move and practice the skill). While these implications are in line with motor learning literature, it should be noted that further evaluation using a later assessment of learning (i.e., a retention or transfer task) is warranted to better understand infant visual motor learning and effective strategies to teach infants motor skills.

### Limitations

There are four main limitations for consideration: sample size, our operational definition of visual anticipation, our use of the Wilcoxon Signed Ranked test, and the lack of a retention or a transfer phase in our experimental design. Given the small sample of classically-defined learners and non-learners, the findings that the two groups are not significantly different, aside from the number of baseline movements, should be replicated in a larger study. This is a first step, exploratory research study as opposed to confirmatory research and a larger sample size is needed. Additionally, this can be further studies through a systematic review utilizing past contingency learning studies. Through utilizing data from past contingency leaning studies, we can further show that the classically-defined learning threshold is insufficient. The controversy surrounding the use of the 1.5× learning threshold to define learners is prevalent in the literature, especially with respect to the development of self-agency [10,11]. Importantly, our findings support the development of self-agency (i.e., predictive gaze behavior) in a group of neurotypical infants who are defined classically as non-learners.

Our definition of visual anticipation has one potential design flaw. Namely, if the infant continues to look at the robot after its activation and then activates the robot again, the infant will be scored as having visual anticipation. One may argue that this is not visual anticipation since the infant is simply staring at a prior location where an event happened, to see if it happens again. While this is a valid argument, in the context of infant cognitive assessments, this is precisely how visual anticipation is assessed. In the Bayley Scales of Infant and Toddler Development 4th edition [21], an assessor identifies if a child has anticipatory gaze by engaging in a game of Peek-a-Boo with the child. The assessor hides from the child and reappears twice in the same location. Then the assessor changes the placement of their head to see if the child assumes that they will move to the prior location. If the child looks or maintains their gaze in the prior region of interest, then they are said to have visual anticipation skills. This example is analogous to our paradigm where the infant is looking at a similar region of interest to see if the robot will activate again.

It could be argued that non-robot looks are time points where the infant is not engaged in the learning paradigm and therefore these non-robot looks should be removed from analysis in order to assesses if the infant is visually anticipating the robot more than random chance (i.e., during the time that the infant is engaged in the paradigm are they visually anticipating the consequences of their actions). If we removed non-robot looks, a t-test that assessed if infants use predictive more than reactive gazes would be better for determining if infants demonstrate anticipatory gaze as opposed to our use of Wilcoxon Signed Ranked test to assess predictive, reactive, and non-robot looks. This argument is valid only if the skill is considered to be already learned. During a learned action, we would expect a performer to be able to visually anticipate the consequences of their actions most of the time if the skill required visual anticipation [35]. However, we contend that during the acquisition of a new skill, we would expect the performer to be exploring multiple behaviors to better understand what behaviors give the desired outcome and what additional behaviors are allowed [8,38,39]. Variability in movement and visual behavior allows for the performer, during the learning process, to explore opportunities that can be used in future situations. For example, non-robot looks can be seen as a chance for infants to explore if socializing with caregiver or another willing participant is allowed in this paradigm. Therefore, we find that the Wilcoxon Signed Ranked test describes and assesses the occurrence of the three types of gazes seen in this paradigm without eliminating any data that could be important for understanding the learning process. However, it should be mentioned that there is no motor literature in the adult and pediatric realm that describes how accurate humans are at visually anticipating a movement that is being learned for the first time, as was the case in our study. This information may be pertinent for understanding how humans use visual behavior to learn novel movements and visual cueing can be used to aid early interventions [40].

Finally, this study did not use an assessment of learning like a transfer task or retention test to assess and classify classically defined learners and non-learners. We used the classically-defined criteria of 1.5 times more movement in contingency compared to baseline within a single session; however, our results suggest that the classic definition of learning a contingency paradigm is insufficient when used within a single exposure session. Further assessment is needed to see how infants demonstrate visual anticipation in a retention or transfer test to see if they are better at predicting the outcomes of their actions once they have practiced those actions. In regard to practicing, we did not ask about use of infant-activated toys in the home. Specifically, the robot in our study was activated by a leg movement of the infant. Anecdotally, these types of toys are more common for older children (e.g., interactive video games) and are still in the exploratory research phase in the age range included here. However, we do note that this is a limitation of the current study and this question should be included in future research.

## 5. Conclusions

In summary, this study presents a novel method to examine the process of infant motor learning in the context of visual motor behavior. We present evidence that, while infants engage in a contingency learning paradigm, they are demonstrating visual anticipation, potentially as evidence of learning the connection between their movement and the outcome for that movement. Additionally, we hypothesize the existence of two learning stages which describe how infants might utilize visual motor behaviors in the process of contingency learning. These learning stages should be further studies in future research.

## Figures and Tables

**Figure 1 sensors-25-00844-f001:**
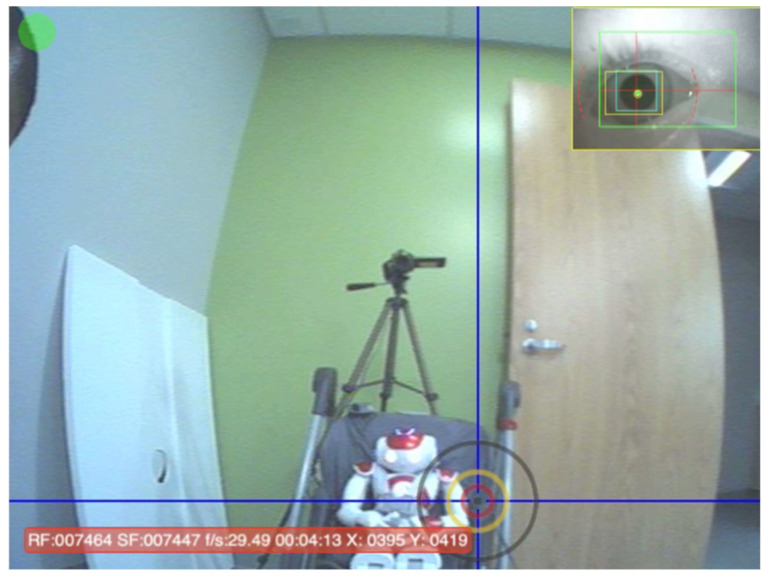
Image of a single frame of the eye gaze data. The most inner red circle is the 4-degree radius.

**Figure 2 sensors-25-00844-f002:**
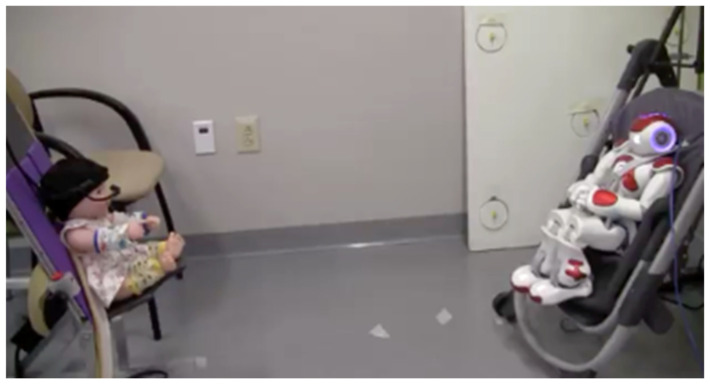
Experimental set up for contingency paradigm.

**Figure 3 sensors-25-00844-f003:**
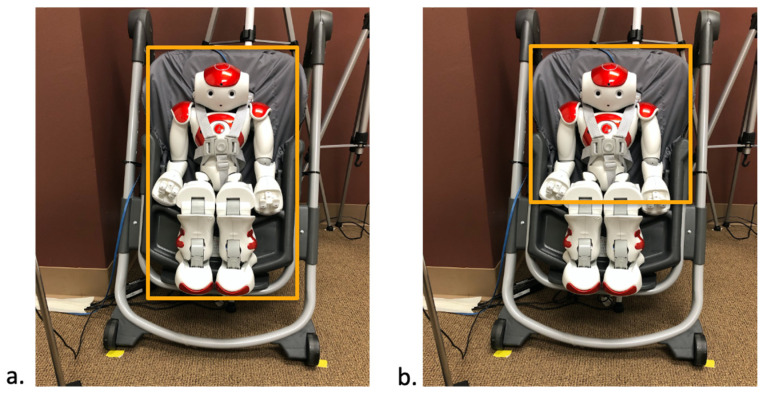
Regions of interest (ROI) for overall looks at the entire robot (**a**) and ROI predictive and reactive looks (**b**). The hands are included in the ROI for (b). Each ROI is boxed in yellow.

**Figure 4 sensors-25-00844-f004:**
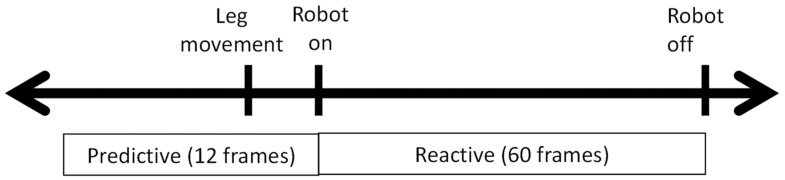
Predictive and reactive gaze coding for a single robot activation. Note: Predictive gaze is defined as occurring within a window of 0.4 s prior to the robot activating, and a reactive gaze fell within a window of 2 s after the robot activated.

**Figure 5 sensors-25-00844-f005:**
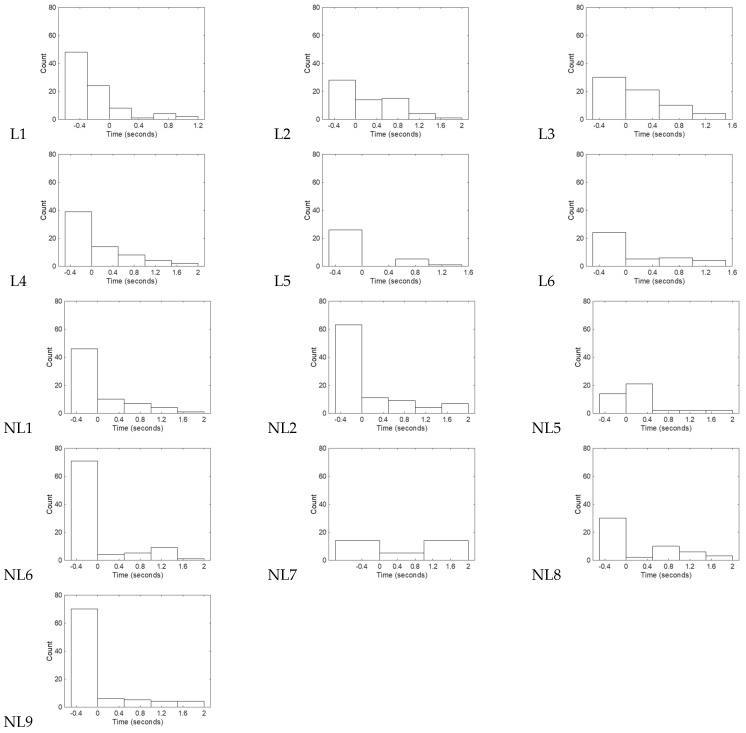
The timing of gazes on the reinforcement for each infant. The x-axis is time in seconds. A value of 0 on the x-axis is when the robot would activate. Negative values on the x-axis mean that the gaze occurred prior to the robot’s activation, while positive values occurred during its activation. The y-axis is the frequency for that block. The dashed line in each plot denoted the median for the distribution of data. Non-robot looks are not included in this analysis.

**Figure 6 sensors-25-00844-f006:**
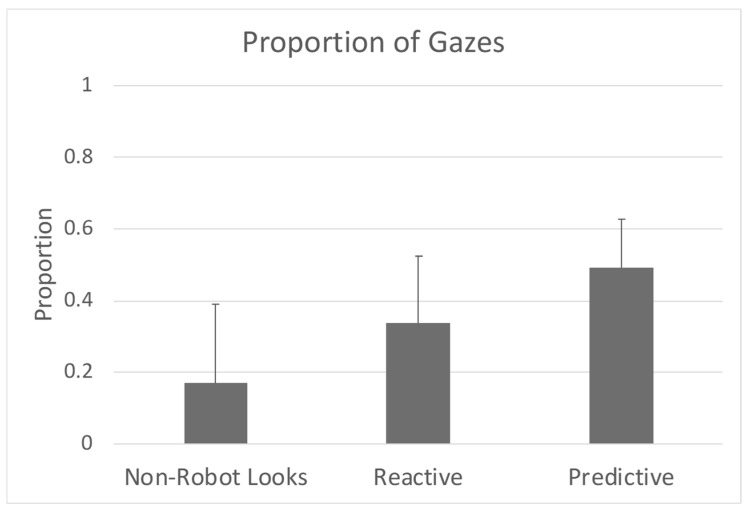
Proportion of gazes for infants classically defined learners. Reactive and predictive gazes are the ROI in Figure 3b. Bars represent the mean for the group and the line is the standard deviation.

**Figure 7 sensors-25-00844-f007:**
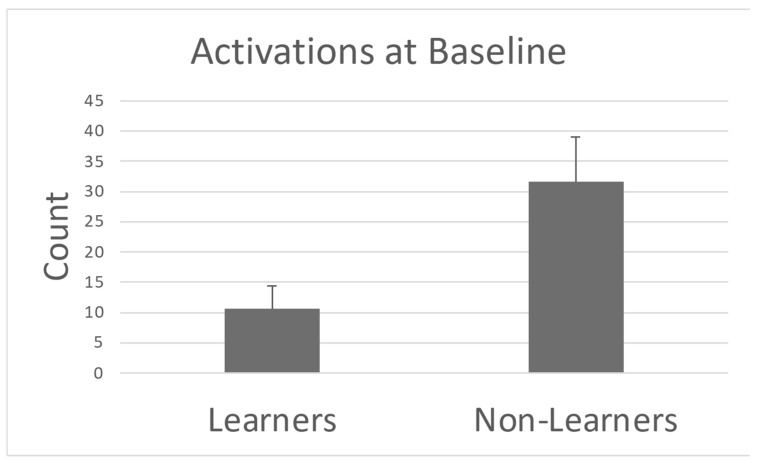
Number of leg movements that would have activated the robot during the baseline for infants classically defined learners and non-learners.

**Table 1 sensors-25-00844-t001:** Participant characteristics for all infants (Mean ± SD).

Variable	Mean (SD)
Age (days)	212 (20)
Sex	7 males, 8 females
Weight (kg)	7.58 (0.59)
Thigh Length (cm)	15.1 (2.0)
Shank Length (cm)	13.7 (1.2)
Thigh Circumference (cm)	23.2 (2.4)
Shank Circumference (cm)	17.4 (1.5)
Foot length (cm)	9.1 (1.0)
Foot width (cm)	4.3 (0.5)
Bayley-4 Cognitive, Percentile *	73 (12)
Bayley-4 Language, Percentile *	66 (17)
Bayley-4 Motor, Percentile *	73 (26)
Ethnicity	6 Hispanic, 9 Not Hispanic/Latino
Race	3 Asian, 3 Black/African American, 2 White, 6 Other/Multi-Racial, 1 Declined to answer

* Bayley-4: Bayley Scales of Infant and Toddler Development, fourth edition.

**Table 2 sensors-25-00844-t002:** Average behavioral state in each condition for all infants. Values are average percent of time spent in each state amongst the group by each phase of the paradigm (Mean (SD)).

Varaible	Baseline			Contingency			Extinction		
State	Alert	Fussy	Crying	Alert	Fussy	Crying	Alert	Fussy	Crying
Percent	92.1 (16.8)	7.3 (15.9)	0.7 (2.6)	95.6 (8.3)	3.5 (6.7)	0.9 (2.4)	94.5 (17.8)	2.7 (7.6)	2.7 (10.6)

**Table 3 sensors-25-00844-t003:** Learning threshold, peak number of robot activations, total number of activations during each contingency condition for each infant.

Infant	Baseline (Number of Activations)	Threshold (Number of Activations)	Peak Contingency Block (Activations/2 min)	Extinction (Number of Activations)	Total Number of Activations
L1	10	15	36	18	100
L2	12	18	28	29	63
L3	16	24	29	20	66
L4	5	7.5	31	22	87
L5	9	13.5	35	9	77
L6	12	18	19	29	41
NL1	40	60	32	31	90
NL2	37	55.5	40	32	124
NL3 *	14	21	13	8	31
NL4 *	42	63	48	47	160
NL5	18	27	16	18	43
NL6	34	51	34	23	110
NL7	26	39	14	34	33
NL8	33	49.5	34	21	72
NL9	33	49.5	41	32	97

* Infants without eye tracker data. Infant classified as classically defined learners are denoted as L and classically defined non-learners as NL.

**Table 4 sensors-25-00844-t004:** Variables for contingency learning paradigm compared between classically defined learners and non-learners. Values in classically defined learners and non-learners columns are median (range). Values in the Statistics column are the Wilcoxon’s W values and *p*-values. Eye gaze variables for classically defined non-learners have an *n* = 7 and are denoted with an asterisk *. Significant results are denoted with two asterisks **.

Variable	Classically Defined Learners (n = 6)	Classically Defined Non-Learners (n = 9)	Statistic (*p*-Value)
Number of leg movements that would have activated the robot during baseline	11 (5–16)	33 (14–42)	W = 98 (*p* = 0.0012) **
Total number of activations	72 (43–100)	90 (31–160)	W = 77 (*p* = 0.607)
Number of leg movements that would have activated the robot during the extinction	21 (9–29)	31 (8–47)	W = 83.5 (*p* = 0.181)
Proportion of predictive gazes	0.46 (0.34–0.71)	* 0.51 (0.33–0.72)	W = 50 (*p* = 1.0)
Proportion of reactive gazes	0.37 (0.08–0.54)	* 0.25 (0.17–0.63)	W = 50 (*p* = 1.0)
Proportion of non-robot looks	0.09 (0.02–0.58)	* 0.18 (0–0.29)	W = 52 (*p* = 0.731)
Skewness for gaze onset on the activation of the robot	1.09 (0.56–2.42)	* 1.69 (0.11–2.4)	W = 53 (*p* = 0.628)
Median onset on the activation of the robot (seconds)	−0.31 (−0.4–0.18)	* −0.34 (−0.40–0.75)	W = 45.5 (*p* = 0.628)
Looking during contingency phase (seconds) (Figure 3a ROI)	267.77 (195.9–277.21)	* 247.83 (152.88–297.00)	W = 44 (*p* = 0.534)
Intertrial duration (seconds)	6.53 (4.76–10.56)	5.31 (2.98–11.89)	W = 69 (*p* = 0.776)

## Data Availability

The data presented in this study are available on request from the corresponding author due to participant privacy.

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
