# Peer review of "Infants Display Anticipatory Gaze During a Motor Contingency Paradigm"

_sensors, 2025, doi:10.3390/s25030844_

Round 1

Reviewer 1 Report

Comments and Suggestions for Authors

The manuscript focuses on an interesting topic: learning motor skills in infants using a contingency paradigm. More exactly, the aim of the study was to determine if infants who learned a contingency visually anticipated the outcomes of their behaviour.

The effort and difficulty involved in conducting research with such young children should be highlighted and positively valued.

Despite the complexity of the topic, the article is clear and understandable. The wording is correct.

Congratulations.

Some issues for improvement related to different sections of the manuscript are indicated below.

1.Introduction

- It is somewhat brief, but generally correct, although it could be improved by explaining in greater depth a relevant issue in the article: the calibration between learners and non-learners and the existing debate about it (lines 83-88). Although the authors of the manuscript indicate references on this issue, they should present this issue more fully since it is a relevant topic for the discussion of their results.

-Lines 88-90. These lines referring to recommendations should be removed from the Introduction section.

2.1. Participants

- Line 97. One point left before Infants.

-Lines 107-110. The explanation about the reasons that reduced the sample from 17 to 15 children should be indicated before (on line 104, right after indicating the target sample size).

- Table 1. The top line of the table is missing. The content of each row should be centered (it is very high). * appears below the table, but * is not inside the table.

2.2. Procedures

- Lines 119-120. The manuscript indicates that infant's anthropometric measurements were obtained. However, these measurements are missing in the manuscript. They should appear at least in Table 1.

- How long did it take to take these measurements and the motor, cognitive and language development assessment? Could this time imply fatigue in the children that affected their subsequent motor learning?

- Line 140. The duration of the contingency paradigm must be indicated precisely: 12 minutes and 10 seconds.

- Line 146. For a higher quality of the manuscript and for it to serve as a learning tool for readers, it would be of interest to explain in more detail, especially, the demonstration and extinction periods. Although it is something basic to the paradigm, including it in the article may be of interest to certain readers.

- Line 147. Figure 1 must be put in colour. In black and white figure, it is difficult to detect the details of interest that the figure tries to highlight.

23. Data Processing

- Line 203.  Figure 2must be put in colour. In black and white figure, it is difficult to detect the details of interest that the figure tries to highlight.

- Line 211. Although the reference is indicated, briefly explaining the behavioral state (sleeping, drowsy, alert, fussy and cryinig) would increase the quality of the manuscript.

- Table 2. Its format should be improved (no vertical lines, cells of the same size, etc.). All behavioral states should appear, even if their statistics are 0.  “Varaible” should be replaced by “variable”.

3. Results

- Table 3. What order has been followed to present the children's data? Why don't all the L children present themselves first and then the NL children?

- Figures: Including the data within the figures would increase the quality of the manuscript.

Author Response

Reviewer 1:  

Some issues for improvement related to different sections of the manuscript are indicated below.

1.Introduction

- It is somewhat brief, but generally correct, although it could be improved by explaining in greater depth a relevant issue in the article: the calibration between learners and non-learners and the existing debate about it (lines 83-88). Although the authors of the manuscript indicate references on this issue, they should present this issue more fully since it is a relevant topic for the discussion of their results.

Response: We added more details to the second paragraph of the introduction to discuss the debate clearer. This is linked to the third paragraph, as well.

-Lines 88-90. These lines referring to recommendations should be removed from the Introduction section.

Response: These lines were removed from the introduction section.

2.1. Participants

- Line 97. One point left before Infants.

Response: the period was added to the recommended area.

-Lines 107-110. The explanation about the reasons that reduced the sample from 17 to 15 children should be indicated before (on line 104, right after indicating the target sample size).

Response: We restructured the two paragraphs as suggested. The paragraph is now ordered as followed: 1) state that no sample size was calculated, 2) states the proposed sample size, and 3) states how the sample of 17 was reduced to 15.

- Table 1. The top line of the table is missing. The content of each row should be centered (it is very high). * appears below the table, but * is not inside the table.

Response: We reformatted table 1. The space between variables was removed to make the table have less space. Second, * now appears in the table next to the Bayley items.

2.2. Procedures

- Lines 119-120. The manuscript indicates that infant's anthropometric measurements were obtained. However, these measurements are missing in the manuscript. They should appear at least in Table 1.

Response: The data requested (thigh and shank length and circumference; and foot length and width) were added to table 1.

- How long did it take to take these measurements and the motor, cognitive and language development assessment? Could this time imply fatigue in the children that affected their subsequent motor learning?

Response: The data collection duration ranged from an hour to an hour and a half. This detail was added to the procedures section. In addition, we added more details about how we prevented infant fatigue by permitting feeding and breaks in between assessments.

- Line 140. The duration of the contingency paradigm must be indicated precisely: 12 minutes and 10 seconds.

Response: We revised the manuscript to state the paradigm was 12 minutes and 10 seconds.

- Line 146. For a higher quality of the manuscript and for it to serve as a learning tool for readers, it would be of interest to explain in more detail, especially, the demonstration and extinction periods. Although it is something basic to the paradigm, including it in the article may be of interest to certain readers.

Response: A paragraph was added to explain the rationale for the chosen structure of the task.

- Line 147. Figure 1 must be put in colour. In black and white figure, it is difficult to detect the details of interest that the figure tries to highlight.

Response: Colour was added to figure 1.

  1. Data Processing

- Line 203.  Figure 2must be put in colour. In black and white figure, it is difficult to detect the details of interest that the figure tries to highlight.

Response: The figure was updated to have a colour image instead of a black and white image.

- Line 211. Although the reference is indicated, briefly explaining the behavioral state (sleeping, drowsy, alert, fussy and cryinig) would increase the quality of the manuscript.

Response: These descriptions were added to work to inform the readers how each behaviour state was defined.

- Table 2. Its format should be improved (no vertical lines, cells of the same size, etc.). All behavioral states should appear, even if their statistics are 0.  “Varaible” should be replaced by “variable”.

Response: Infants were not assessed during a drowsy or sleeping state. If an infant arrived to the assessment asleep, they were allowed to sleep and the caregiver filled out paperwork that did not require the infants participation (i.e. filling out the informed consent, the FYI, or answering demographic questions). We added the information that infants were not assessed when drowsy or sleeping. This allows for a more condensed table. 

  1. Results

- Table 3. What order has been followed to present the children's data? Why don't all the L children present themselves first and then the NL children?

Response: Table 3 was restructured to present all L first and all NL after.

- Figures: Including the data within the figures would increase the quality of the manuscript.

Response: We request that we are provided with the specific figure this comment is addressing and what specific data needs to be included (e.g. individual data).

Reviewer 2 Report

Comments and Suggestions for Authors Abstract:

Is the division of the abstract into three sections really necessary? I suggest simplifying the format into a more condensed text that better conveys the main objectives, methodology, and findings of the study.

Introduction:

  • Please provide specific references when discussing studies individually, e.g., [1], [2], [3], [4]. A similar approach should be applied for [5]-[7], [12]-[13], and others where applicable. Repeated citations are unnecessary and make the text harder to follow.

  • The statement: “Our primary aim was to determine if infants who were classified as learners (using the classical method)” requires clarification. What method was used? The lack of reference to a detailed description makes it difficult to understand how infants were classified as learners.

  • Citations: Chains like [14], [15], [20], [25], [26] are imprecise. Please clearly indicate which parts of the article the authors are referencing.

  • Hypotheses: They should be written in separate lines for better readability. Could the hypotheses be formulated more clearly? Simplification is suggested.

  • The phrase: “Based on our findings, we offer...” should be changed to “based on the results of our study...”.

Participants:

  • The study group consisted of only 15 children, which is insufficient. Such a small sample size makes it impossible to obtain reliable data distributions. Under current standards, at least 100 children should be included in the study, with at least 40-50 in each experimental group (which is still minimal!).

  • The argument about limitations due to COVID-19 is no longer valid.

  • References to positions [10] (2014), [28] (2002), and [29] (1980) are outdated. Please refer to current standards (e.g., Infancy, 2025; 30:e12633).

  • Additional questions:

    • Was the recruitment paid?

    • Were factors like therapies, rehabilitation, and the presence of devices activated by the child at home considered? These could significantly affect the results.

    • Was it checked whether the children had strabismus? Eye alignment can influence eye-tracking results. Different eye colors can also affect tracking efficiency, as demonstrated in studies on newborns.

  • The description of Hispanic/non-Hispanic in combination with racial categorization seems unnecessary in this form.

FYI Data:

The statement: “Data from the FYI are not reported here; instead they were used for a companion study...”. Please ensure access to these data in an open repository, as they are crucial for the results.

Methods:

  • Infant chair: A diagram, drawing, or photo is essential to enable replication of the study.

  • Eye-tracking:

    • The Positive Science eye tracker tracks only one eye. Was the right and left eye of the children alternately examined? If not, another control group is necessary.

    • Was the dominant eye of the child determined?

    • Calibration results: What level of accuracy was accepted? An average value is insufficient—outliers must be excluded.

  • Experimental conditions:

    • Was the laboratory completely isolated from external sounds?

    • Was a parent accompanying the child? If so, where were they seated?

    • Where was the researcher positioned?

ROI (Regions of Interest):

  • Why does the reactive ROI not include the robot's hands? Please explain how overlapping data between ROIs was interpreted.

  • I suggest more intuitive names, such as “entire robot” or “reward areas”.

  • Were the robot’s lights colorful? If so, could all children perceive the colors?

Results:

  • Figure 3: Please clarify and simplify the visualization.

  • ROIs should be dynamic and include only the active reward areas.

  • It is not possible to reliably comment on results derived from such a small sample size.

Summary: The study has significant methodological limitations:

  1. The sample size is too small (15 participants), making reliable statistical analyses impossible.

  2. Outdated references and a lack of precision in citations.

  3. Insufficient description of the study conditions, including equipment, laboratory setup, and procedures.

  4. Questions about ROI and the interpretation of eye-tracking results.

I recommend rejecting the article in its current form. In the future, the authors should:

  • Increase the sample size (a minimum of 100 children is essential) and analyze whether the results follow a normal distribution.

  • Provide a better description of the methodology, including a detailed explanation of the equipment and procedures.

  • Consider using more advanced eye-tracking equipment (at least 60Hz + non-head-mounted ET systems).

Best of luck in future research.

Author Response

Reviewer 2:

Abstract:

Is the division of the abstract into three sections really necessary? I suggest simplifying the format into a more condensed text that better conveys the main objectives, methodology, and findings of the study.

Response: The division of submitted abstract has 4 sections: Background, Methods, Results, Conclusions (not three as stated in the comment).  These divisions for the abstract are according to Sensors’ instructions for the author (see Front Matter in the following link: https://www.mdpi.com/journal/sensors/instructions). The abstract was not changed according to the feedback in the revised submission, since it goes against the Journal’s instruction. We are willing to revise the abstract’s divisions at the direction of the editor and/or Journals instructions.

Introduction:

  • Please provide specific references when discussing studies individually, e.g., [1], [2], [3], [4]. A similar approach should be applied for [5]-[7], [12]-[13], and others where applicable. Repeated citations are unnecessary and make the text harder to follow.

Response: We mentioned specific aspects of the studies references in this comment and restructured accordingly. The grouping presented in the revised manuscript are adjusted as appropriate.

  • The statement: “Our primary aim was to determine if infants who were classified as learners (using the classical method)” requires clarification. What method was used? The lack of reference to a detailed description makes it difficult to understand how infants were classified as learners.

Response: The classical method was defined and discussed in the second paragraph of the introduction and in section 2.4.1 in the methods. We added a parentheses statement next to the line in mention to direct the reader to the section where this was stated prior and operationally defined in the methods.

  • Citations: Chains like [14], [15], [20], [25], [26] are imprecise. Please clearly indicate which parts of the article the authors are referencing.

Response: This chain of articles was rewritten into paragraphs 2 and 3 of the introductions to reduce the clutter and state the specific articles more precisely.

  • Hypotheses: They should be written in separate lines for better readability. Could the hypotheses be formulated more clearly? Simplification is suggested.

Response: Recent articles published in sensors (see https://doi.org/10.3390/s24237586) format the hypotheses as presented in the submitted work. In keeping with standard publishing guidelines, we choose to keep the document as written. We are open to adjusting per direction by the editing staff.

  • The phrase: “Based on our findings, we offer...” should be changed to “based on the results of our study...”.

Response: The phrase in mention was deleted in response to reviewer 1’s comments as it is a discussion point, not an introductory point.

Participants:

  • The study group consisted of only 15 children, which is insufficient. Such a small sample size makes it impossible to obtain reliable data distributions. Under current standards, at least 100 children should be included in the study, with at least 40-50 in each experimental group (which is still minimal!).

Response: The sample size was chosen based on prior literature using the contingency learning paradigm, where significant effects are observed. This is an exploratory, first step research study which makes determining a specific sample size difficult as there are no preliminary or similar studies that provide effect sizes. While enrolling the largest sample size possible would give us the most power to detect small effects we are, unfortunately, limited by available resources. It takes a lot of time and effort to recruit infant participants for human subjects research. Further, it takes resources to collect the data (space, equipment, personnel time, money). Resources are not unlimited and so we have to make a decision that allows us to collect a sample that is “large enough” to be useful as a first step of research exploratory study, while balancing this against available resources. As such, our goal here is to provide useful exploratory research.  In this regard, Davis-Kean & Ellis (2019) emphasize the importance of delineating between confirming and exploratory research. Following these guidelines, we reaffirm that the primary goal of this study is exploratory, not confirmatory. We have added this statement at line 110 and in the limitations section at line 546. Given that this is an exploratory study and that the sample size is consistent with what has been published prior, we advocate for the value of our work to lay a foundation necessary to support future, larger studies.

  • The argument about limitations due to COVID-19 is no longer valid.

Response: We removed the reference to COVID-19.

  • References to positions [10] (2014), [28] (2002), and [29] (1980) are outdated. Please refer to current standards (e.g., Infancy, 2025; 30:e12633).

Response: We appreciate that you recommended a piece of eye tracking literature (Infancy, 2025; 30:e12633; DOI: https://doi.org/10.1111/infa.12633), however the paper uses a completely different paradigm from the work being presented in our paper. The paper you suggest uses the “face pop-out task”. This task only requires the infant to look at items presented on the screen and reports the duration of time spent looking at faces. Our paradigm is assessing if infants visually anticipate that they will activate the robot. In our experimental paradigm, the infant learns that their movement activates the robot in front of them. We are unaware of a paper that describes its content as a standard for eye tracking paradigms but are happy to include such a paper if it exists and we have missed it.

 The references cited in our paper are example of the historical literature and prior example of the learning threshold for this paradigm. All these papers mentioned cite an experimental set up where the infant is in control of the stimulus and activate it the stimulus with their movement. We cited an article from 2023 and removed the other references to provide the readers with a current example that cites the former references.

  • Additional questions:
    • Was the recruitment paid?

Response: Enrolled participants received $40 (USD) for their participation in the study. This detail was added to the paper.

    • Were factors like therapies, rehabilitation, and the presence of devices activated by the child at home considered? These could significantly affect the results.

Response: As stated in our participants section, these children were born with typical development. Children with any diagnoses that would have qualified them to receive rehabilitation or early intervention therapies were excluded from the study. We did not ask about the presence of devices activated by the child at home and have added this to the limitations section at line 601.

    • Was it checked whether the children had strabismus? Eye alignment can influence eye-tracking results. Different eye colors can also affect tracking efficiency, as demonstrated in studies on newborns.

Response: In our exclusion criteria, children were excluded if they had visual impairments or eye conditions. While Strabismus was not assessed, these children would have been excluded if they had this condition among other conditions. As for your comment on eye color affecting the tracking efficiency. There is no evidence to support that the positive science eye tracker used in the study records different eye colors at different efficiencies. We present our eye tracking data to the standards that have been published prior using the positive science head mounted eye tracker. Last, the paper you referred to as the “current standard” (Infancy, 2025; 30:e12633; DOI: https://doi.org/10.1111/infa.12633) does not make reference for checking for strabismus.

  • The description of Hispanic/non-Hispanic in combination with racial categorization seems unnecessary in this form.

Response: The area that these data were collected in has a Hispanic population of approximately 48%. Therefor it is important to describe that we are collecting a sample that is close to the populations ethnic and racial makeup. These data are relevant to the work.

FYI Data:

The statement: “Data from the FYI are not reported here; instead they were used for a companion study...”. Please ensure access to these data in an open repository, as they are crucial for the results.

Response: The FYI in this work was used a measure for autistic traits. The data being presented is from the cohort of children who are not at elevated likelihood for developing autism. As stated in the data availability section: “The data presented in this study are available on request from the corresponding author due to participant privacy”. The results from the companion study can also be found Marcelo Rosales’ dissertation, available on the University of Southern California’s library website. We cited Dr. Rosales’ dissertation for these data in the revisions.  

Methods:

  • Infant chair: A diagram, drawing, or photo is essential to enable replication of the study.

Response: Figure 2 of the experimental setup was added to the manuscript.

  • Eye-tracking:
    • The Positive Science eye tracker tracks only one eye. Was the right and left eye of the children alternately examined? If not, another control group is necessary.

Response: The right eye for all participants was tracked using the positive science eye tracker. This detail was added to the revised document. As to your point about the need for a control group using the other eye, please cite explain your rational for this statement. Our work uses the standard practices for the positive science eye tracking tool, and no publication to date using the positive science eye tracker controls for the contralateral eye. In addition, these are infants ages to 6 to 9 months with typical developing vision, and there is no reason to suspect atypical vision. Please explain your rational for expecting drastic differences between the eyes.

    • Was the dominant eye of the child determined?

Response: As stated in our previous response, these infants are typically developing, and we would expect no differences between the eyes. In addition, the common test for eye dominance like the Miles test or Porta test, to our knowledge, cannot be performed in our age range (6-9 months). There is work to suggest that eye dominance is correlated with hand preference, however, there is mixed results regarding this statement. In addition, these infants are 6 to 9 months. Durning this time, hand preference is fluid and not established. Therefore, it can be hypothesized that eye dominance at this age would not be established. Last, the paper you cited previously as the “current standard” (Infancy, 2025; 30:e12633; DOI:https://doi.org/10.1111/infa.12633), does not determine eye dominance or factor eye dominance into their model. We ask that you present empirical literature showing why eye dominance would factor into our results and why eye dominance would be established at this age.

    • Calibration results: What level of accuracy was accepted? An average value is insufficient—outliers must be excluded.

Response: The acceptable level of accuracy of a 4-degree radius and none of the participants in this data set exceeded this amount. These data were added to the revised document. In addition, the way these data are presented are in line with the standard practice for the positive science eye tracker (see https://doi.org/10.1111/desc.12626).

  • Experimental conditions:
    • Was the laboratory completely isolated from external sounds? Was a parent accompanying the child? If so, where were they seated? Where was the researcher positioned?

Response: We added these details to the revised manuscript. The laboratory was isolated from external sounds, the parent sat to the left of the child, and the researcher sat behind the child.

ROI (Regions of Interest):

  • Why does the reactive ROI not include the robot's hands? Please explain how overlapping data between ROIs was interpreted.

Response: The ROI does include the robot’s hands. The robot would bring its hands to midline and the clapping would occur near the chest of the robot. Figure 3 was edited to bring the line down so that the hands are in view for the reader. In addition, as requested in a later comment, we added a video file that shows the ROI during the robot’s activation.

  • I suggest more intuitive names, such as “entire robot” or “reward areas”.

Response: We added the language, the entire robot and any part of the robot for Figure 3 a. The reward area is an inaccurate description of the robot activations because it cannot be determined if the infants see activation of the robot as a reward. While the infant might show presumed signs of enjoyment, these are all assumptions. This why the field has move toward saying reinforcement, rather than reward. To be objective, we refrain from using the “reward area” in our work.

  • Were the robot’s lights colorful? If so, could all children perceive the colors?

Response: We added the detail that the robot displayed colorful lights. As stated in our responses and in our manuscript, our study population was infants ages 6 to 9 months. Given that infants this age are non-verbal, there is no way to assess if they perceive colors.

Results:

  • Figure 3: Please clarify and simplify the visualization.

Response: We added more details for figure 3, to aid the readers in interpreting the figure. Figure 3 is a single robot activation where the infant activated the robot with a leg movement, the robot turns on, and then the robot turns off. Predictive and reactive gazes occur in the window below the timeline as depicted and described.

  • ROIs should be dynamic and include only the active reward areas.

Response: We included a video representation of the ROI in a supplemental figure.

  • It is not possible to reliably comment on results derived from such a small sample size.

Response: As stated in our response to the prior comment, the sample size was determined using prior literature. Additionally, for this novel methodology and proof of concept, we find that the sample size is sufficient. Davis & Ellis (2019) emphasize the importance of delineating between confirming and exploratory research (see https://doi.org/10.1016/j.infbeh.2019.101339). Following these guidelines, we reaffirm that the primary goal of this study is exploratory; to uncover and demonstrate the diverse developmental trajectories in infants. We have added this information at line 110 and in the limitations section at line 546.

Summary: The study has significant methodological limitations:

  1. The sample size is too small (15 participants), making reliable statistical analyses impossible.
  2. Outdated references and a lack of precision in citations.
  3. Insufficient description of the study conditions, including equipment, laboratory setup, and procedures.
  4. Questions about ROI and the interpretation of eye-tracking results.

I recommend rejecting the article in its current form. In the future, the authors should:

  • Increase the sample size (a minimum of 100 children is essential) and analyze whether the results follow a normal distribution.
  • Provide a better description of the methodology, including a detailed explanation of the equipment and procedures.
  • Consider using more advanced eye-tracking equipment (at least 60Hz + non-head-mounted ET systems).

Best of luck in future research.

Response to Summary: Most of these points have been addressed in our repones above.  The points addressed above include: the justification for our sample size, discussion of how our references are current and relevant, the revisions have sufficient descriptions of the study conditions and methods, and the issues regarding the ROI have been addressed.

In response to your comment regarding the eye tracker used for this study. The positive science head mounted eye tracker is a valid tool and is cited in rigorous scientific literature. This article presents the standard practices for this device and do not go beyond the tools’ limitations. The specific methodological comments regarding the eye tracker and the study population have been addressed accordingly.

Round 2

Reviewer 2 Report

Comments and Suggestions for Authors

Thank you to the Authors for revising the introduction and implementing relevant changes in the discussion. However, not all responses are satisfactory.

Statistical analysis – The provided statistical analyses are unsatisfactory. There is no indication of statistical power or the distribution of measurement points for individual children (e.g., box plots with dots or box plots combined with violin plots).

Sample size – The groups are alarmingly small (6 and 9 participants, effectively 7 due to the lack of ET registration for 2 participants). This must be clearly indicated or, ideally, the sample size should be expanded. Using statistical analyses, please specify what target group the data suggest. In its current form, I cannot accept this aspect of the manuscript.

Data anonymization and availability – Please anonymize the data (e.g., assign numbers as identifiers in the article, blur the faces of parents and researchers in the videos) and provide the data in an open repository. This aligns with the principles of open science and data transparency. As a reviewer, I have no way to fully verify your results and data. Additionally, since this is a double-blind review, I cannot share my email address to request further clarification.

Figure 6. Proportion of gazes for infants classically defined learners – I do not understand this figure. What is defined as "1"? Is it the total number of gazes? Additionally, based on the designated ROI, the reaction area appears to overlap. This overlap may have also influenced the lack of statistical deviations in the ET data for both subgroups presented in Table 4.

"Seventeen infants … received $40 (USD) for their participation" – It seems unlikely that the infants themselves received the payment; this must have been provided to their parents. Please clarify.

The division into ROIs raises concerns, as it directly impacts the interpretation of results. Although the ROI was modified in the illustration, the data presented were not updated to reflect this change. This requires clarification.

Author Response

Statistical analysis – The provided statistical analyses are unsatisfactory. There is no indication of statistical power or the distribution of measurement points for individual children (e.g., box plots with dots or box plots combined with violin plots).

Response: We request that you justify the specific areas that are unsatisfactory. We present the appropriate statistical analysis for these data and justify each decision for the statistics provided in the work. In addition, in line with research manuscript presentations, we use effect sizes and p-values to strength our work (see: Sullivan and Feinn 2012. doi: 10.4300/JGME-D-12-00156.1). Effect size and p-values are more appropriate for manuscript reporting.

Power is not appropriate for every research study; this is likely why Sensors does not require it. In addition, as Dorey 2010 (doi: 10.1007/s11999-010-1435-0) points out in their discussion about power calculation, calculating power after data have been collected does not change the fact that novel and relevant findings are being presented. In addition, power is a variable that is often defined by researcher during data planning and is often adjusted to make research planning more feasible. For example, if a power of 0.9 produced an unrealistic sample size, then a lower power might be chosen for the proposed research. Given that this is an exploratory research study, the data presented here would be used in our future grant proposals for confirming studies and we will utilize the variable of power then. As we explained earlier; given this is a novel approach there were not pre-existing data from which to calculate power before conducting the study. And we have explained above the limitations of calculating power after a study has been conducted. We support that effect sizes and p-values are more appropriate here, as we reported.

In summary, we present appropriate statistical analysis, support our statistics using effect sizes according to literature, and will use power in the future. We request information regarding the “unsatisfactory” analysis you are claiming. Please cite literature when appropriate, and support broad claims. Regarding the individual data, we have, and still present individual anonymize data when appropriate. The two supplemental figures, Table 3, and Figure 5 all present individual data.

Sample size – The groups are alarmingly small (6 and 9 participants, effectively 7 due to the lack of ET registration for 2 participants). This must be clearly indicated or, ideally, the sample size should be expanded. Using statistical analyses, please specify what target group the data suggest. In its current form, I cannot accept this aspect of the manuscript.

Response: As stated in our prior response and in the paper “This is a first step, exploratory research study (as opposed to confirmatory research).” We mention our reasoning for the sample sized collected and acknowledge that the size is a limiting factor. In addition, we also state that replication with a larger study should be performed to examine our proposed learning stages and findings in a confirming study.

In addition, we recommend examining the specific layers to this paper. While the comparisons groups are small, the effect sizes that show significant differences are large and support the claim “learners and non-learners are more similar when they are engaged with learning the connection between their actions and the robot”. Expanding upon this, we show the individual data in the supplements and manuscripts to further show the similarities of the whole sample. In addition, these data are anonymized, and you can track individual participants as requested in your following comment. The data from the whole sample shows similar trends in the behavior that are stated in our discussion and conclusion. We present future hypothesis and state the need for replication with a larger sample size. Please see line 542 in the revised manuscript, where we have added an additional mention of the limitation of the small sample size, in addition to the already present discussions of sample size.

Data anonymization and availability – Please anonymize the data (e.g., assign numbers as identifiers in the article, blur the faces of parents and researchers in the videos) and provide the data in an open repository. This aligns with the principles of open science and data transparency. As a reviewer, I have no way to fully verify your results and data. Additionally, since this is a double-blind review, I cannot share my email address to request further clarification.

Response: In our responses, we have never jeopardized the integratory of the double-blind review and have never, and still do not request contact information. As pointed out in our responses above, we present anonymized data when appropriate in the two supplemental figures, Table 3, and Figure 5. All these figures use the same anonymized identifiers throughout the article and have remained untouched through the revisions process. We request that you specify specific areas where more individual data is needed and justify the rationale for these requests.

In response to the broad and general request for all anonymized data (especially images of participants who are minors), we request justification for the data that is far beyond what is presented in this comment. We are strongly concerned with this comment. Especially the wording where it is quoted to state the need “to fully verify” the work. As stated by Harvard Kennedy School (https://journalistsresource.org/media/peer-review-research-journalists/#:~:text=1.,validate%20research%2C%20not%20verify%20it.), the job of a reviewer is to validate, not verify a piece of work. The job of a reviewer is reviewing the work in its entirety and comment on higher order concepts like literary background, methods, statistical analysis, discussion points, and conclusions. Sensors does not require the data be presented in an open repository and we request a specific rationale as to why it is necessary that our data be in an open repository.

Figure 6. Proportion of gazes for infants classically defined learners – I do not understand this figure. What is defined as "1"? Is it the total number of gazes? Additionally, based on the designated ROI, the reaction area appears to overlap. This overlap may have also influenced the lack of statistical deviations in the ET data for both subgroups presented in Table 4.

Response: The definition for Proportion of gaze is clearly defined in section 2.4.3. It is “The total number of predictive, reactive, and non-robot looks were divided by the total number of times the infant activated the robot. This was performed to calculate the proportion of responses for each type of gaze.”

In response to the overlap comment, the variables for each ROI are different. Please review section 2.3. More details are provided in response to the last comment. However, these data remain unchanged. Please elaborate on your rationale for suspecting statistical deviations in detail and using literary citations when appropriate.   

"Seventeen infants … received $40 (USD) for their participation" – It seems unlikely that the infants themselves received the payment; this must have been provided to their parents. Please clarify.

Response: The phrase was modified to say the caregivers received $40 for their infant’s participation.

The division into ROIs raises concerns, as it directly impacts the interpretation of results. Although the ROI was modified in the illustration, the data presented were not updated to reflect this change. This requires clarification.

Response: Unlike a desk mounted eye tracker, which requires computer programing to define the ROI and process the data, our eye tracking data uses what is called behavioral coding or a frame-by-frame analysis of the data. Trained behavioral coders annotate all the times the infants look in the specified regions and their reliability is checked through double coding the files. All gaze variables had a reliability above 87.5%. We added more details that behavioral coders were given in addition to the depictions.

The update of the depiction for the ROI was done at the request of the reviewer because it was unclear if the hands were contained in the ROI. We added the detail that the hands are included in the ROI for the revised paper. In addition, we reinserted the original ROI depiction with a thinner outline instead of the requested ROI from reviewer 2. We also added more information to Figure 3 to state that hands were included in the ROI. All data presented in the manuscript are in line with the submitted ROI and description. Last, the gif file of the ROI, requested by reviewer 2, was introduced in the prior revisions.

Round 3

Reviewer 2 Report

Comments and Suggestions for Authors

Thank you for the revisions made so far. I would like to share my final comments and observations to further refine the article:

  1. Figure 6: Proportion of Gazes for Infants - Classically Defined LearnersThe caption should clearly explain what the data represents, particularly specifying which ROIs (Regions of Interest) are being referred to. If two ROIs overlap, the method of classification should be clarified. For example, how was the proportional distribution handled if the ROIs overlapped? This is especially important as the sum of the three categories might not equal 1 in such cases.
  2. It might also be helpful to include an additional bar representing data loss (e.g., for closed eyes or blinks).
  3. Anonymized data from the study should ideally be made available in an open repository to ensure transparency and reproducibility. However, I defer to the Editor’s decision (!) on whether the journal’s standards require this for publication.

These are my last comments on the manuscript. Please consider the suggestions provided and incorporate them as needed.

Author Response

1. Figure 6: Proportion of Gazes for Infants - Classically Defined Learners The caption should clearly explain what the data represents, particularly specifying which ROIs (Regions of Interest) are being referred to. If two ROIs overlap, the method of classification should be clarified. For example, how was the proportional distribution handled if the ROIs overlapped? This is especially important as the sum of the three categories might not equal 1 in such cases.

Response: We added more details to explicitly state that Figure 3a was used to calculate the total time spent looking at the robot (lines 230-231, and 290-292). The ROI in Figure 3b as stated in section 2.3 and 2.4.2 is used for predictive and reactive gaze. In Summary, while the two ROI’s overlap, they are used for separate variables. Figure 6 only used the ROI in Figure 3b. The sum does equal 1 for each individual participant. Details were added to the figure to explicitly say that these are the means and standard deviation for the three types of gazes for Classically defined learners.

2. It might also be helpful to include an additional bar representing data loss (e.g., for closed eyes or blinks).

Response: We included these data in the manuscript (see section 2.3). Prior papers published using the positive science eye tracker do not report the amount of data loss due to blinking or closed eyes. See Franchak et al. 2017; https://doi.org/10.1111/desc.12626 and Fears et al. 2019 https://doi.org/10.1093/ptj/pzz027. However, to report high quality rigorous research we added these data since they are reported with desk mounted eye tracking.  

3. Anonymized data from the study should ideally be made available in an open repository to ensure transparency and reproducibility. However, I defer to the Editor’s decision (!) on whether the journal’s standards require this for publication.

Response: In all iterations of the manuscript, we have provided anonymized data. We defer to the editor to decide on whether more anonymized data is needed and if an open repository is necessary.